# TRUTHFUL SELF-PLAY

**Shohei Ohsawa**
Founder & CEO
Daisy AI
6-13-9 Ginza, Chuo-ku, Tokyo, Japan
`o@daisy.inc`

## ABSTRACT

We present a general framework for evolutionary learning to emergent unbiased state representation without any supervision. Evolutionary frameworks such as self-play converge to bad local optima in case of multi-agent reinforcement learning in non-cooperative partially observable environments with communication due to information asymmetry. Our proposed framework is a simple modification of self-play inspired by mechanism design, also known as *reverse game theory*, to elicit truthful signals and make the agents cooperative. The key idea is to add imaginary rewards using the peer prediction method, i.e., a mechanism for evaluating the validity of information exchanged between agents in a decentralized environment. Numerical experiments with predator prey, traffic junction and StarCraft tasks demonstrate that the state-of-the-art performance of our framework.

## 1 INTRODUCTION

Evolving culture prevents deep neural networks from falling into bad local optima (Bengio, 2012). Self-play (Samuel, 1967; Tesauro, 1995) has not only demonstrated the ability to abstract high-dimensional state spaces as typified by AlphaGo (Silver et al., 2017), but also improved exploration coverage in partially observable environments. Communication (Sukhbaatar et al., 2016; Singh et al., 2019) exchanges their internal representations such as explored observation and hidden state in RNNs. Evolutionary learning is expected to be a general framework for creating superhuman AIs as such learning can generate a high-level abstract representation without any bias in supervision.

However, when applying evolutionary learning to a partially observable environment with non-cooperative agents, improper bias is injected into the state representation. This bias originates from the environment. A partially observable environment with non-cooperative agents induces actions that disable an agent from honestly sharing the correct internal state resulting in the agent taking actions such as concealing information and deceiving other agents at equilibrium (Singh et al., 2019). The problem arises because the agent cannot fully observe the state of the environment, and thus, it does not have sufficient knowledge to verify the information provided by other agents. Furthermore, neural networks are vulnerable to adversarial examples (Szegedy et al., 2014) and are likely to induce erroneous behavior with small perturbations. Many discriminative models for information accuracy are available; these include GANs (Goodfellow et al., 2014; Radford et al., 2016) and curriculum learning (Lowe et al., 2020). However, these models assume that accurate samples can be obtained by supervision. Because of this assumption, is it impossible to apply these models to a partially observable environment, where the distribution is not stable.

We generalize self-play to non-cooperative partially observable environments via mechanism design (Myerson, 1983; Miller et al., 2005), which is also known as *reverse game theory*. The key idea is to add imaginary rewards by using the peer prediction method (Miller et al., 2005), that is, a mechanism for evaluating the validity of information exchanged between agents in a decentralized environment, which is calculated based on social influence on the signals. We formulate the non-cooperative partially observable environment as an extention of the *partially observable stochastic games* (POSG) (Hansen et al., 2004); introduce truthfulness (Vickrey, 1961), which is an indicator of the validity of state representation. We show that the imaginary reward enables us to reflect the bias of state representation on the gradient without oracles.

As the first contribution, we propose *truthful self-play* (TSP) and analytically demonstrate convergence to the global optimum (Section 4). We propose the imaginary reward on the basis of the peer prediction method (Miller et al., 2005) and apply it to self-play. The mechanism affects the gradient of the local optima, but not the global optima. The trick is to use *the actions taken by the agents* as feedback to verify the received signal from the every other agent, instead of the true state, input, and intent, which the agents cannot fully observe. TSP only requires a modification of the baseline function for self-play; it drastically improves the convergence to the global optimum in Comm-POSG.

As the second contribution, based on the results of numerical experiments, we report that the TSP achieved state-of-the-art performance for various multi-agent tasks made of up to 20 agents (Section 5). Using predator prey (Barrett et al., 2011), traffic junction (Sukhbaatar et al., 2016; Singh et al., 2019), and StarCraft (Synnaeve et al., 2016) environments, which are typically used in Comm-POSG research, we compared the performances of TSP with the current neural nets, including the state-of-the-art method, with LSTM, CommNet (Sukhbaatar et al., 2016), and IC3Net (Singh et al., 2019). We report that the model with IC3Net optimized by TSP has the best performance.

This work is the first attempt to apply mechanism design to evolutionary learning. TSP is a general optimization algorithm whose convergence is theoretically guaranteed for arbitrary policies and environments. Since no supervision is required, TSP has a wide range of applications to not only game AIs (Silver et al., 2017), but also the robots (Jaderberg et al., 2018), chatbots (Gupta et al., 2019; Chevalier et al., 2019), and autonomous cars (Tang, 2019) employed in multiagent tasks.

**Notation:** Vectors are columns. Let $[\![n]\!] := \{1, \ldots, n\}$. $\mathbb{R}$ is a set of real numbers. $\boldsymbol{i}$ is the imaginary unit. $\operatorname{Re} u$ and $\operatorname{Im} u$ are a real and an imaginary part of complex number $u$, respectively. $n$-tuple are written as boldface of the original variables $\mathbf{a} := \langle a_1, \ldots, a_n \rangle$, and $\mathbf{a}_{-i}$ is a $(n-1)$-tuple obtained by removing the $i$-th entry from $\mathbf{a}$. Let $\mathbf{1} := (1, \ldots, 1)^{\mathrm{T}}$. Matrices are shown in uppercase letters $\mathcal{L} := (\ell_{ij})$. $E$ is the unit matrix. The set of probability distributions based on the support $\mathcal{X}$ is described as $\mathfrak{P}(\mathcal{X})$.

## 2   RELATED WORK

Neural communication has gained attention in the field of *multiagent reinforcement learning* (MARL) for both discrete (Foerster et al., 2016) and continuous (Sukhbaatar et al., 2016; Singh et al., 2019) signals. Those networks are trained via self-play to exchange the internal state of the environment stored in the working memory of recurrent neural networks (RNNs) to learn the right policy in partially observable environments.

The term self-play was coined by the game AI community in the latter half of the century. Samuel (Samuel, 1967) introduced self-play as a framework for sharing a state-action value among two opposing agents to efficiently search the state space at Checkers. TD-Gammon (Tesauro, 1995) introduced self-play as a framework to learn $\mathrm{TD}(\lambda)$ (Sutton & Barto, 1998) and achieve professional-grade levels in backgammon. AlphaGo (Silver et al., 2017) defeated the Go champion by combining supervised learning with professional game records and self-play. AlphaZero (Silver et al., 2018) successfully learnt beyond its own performance entirely based on self-play. All these studies explain that eliminating the bias of human knowledge in supervision is the advantage of self-play.

Self-play is also known as *evolutionary learning* (Bengio, 2012) in the deep learning community mainly as an approach to emerging representations without supervision (Bansal et al., 2018; Balduzzi et al., 2019). Bansal et al. (2018) show that competitive environments contribute to emerging diversity and complexity. Rich generative models such as GANs (Goodfellow et al., 2014; Radford et al., 2016) are frameworks for acquiring an environmental model by employing competitive settings. RNNs such as world models (Ha & Schmidhuber, 2018; Eslami et al., 2018) are capable of more comprehensive ranges of exploration in partially observable environments and generation of symbols and languages (Bengio, 2017; Gupta et al., 2019; Chevalier et al., 2019). The difference between evolutionary learning and supervised learning is the absence of human knowledge and oracles.

Several works have formalized those in which the agents exchange environmental information as a formal class of the games such as Dec-POMDP-Com (Goldman & Zilberstein, 2003) and COM-MTDP (Pynadath & Tambe, 2002), and several frameworks are proposed to aim to solve the problems. However, the limitation of the frameworks is that they assume a common reward. As there are yet no formal definition of *non-cooperative* communication game, we formalize such a game to

Comm-POSG as a superset of POSGs (Hansen et al., 2004), a more general class of multi-agent games including the cases of non-cooperativity (Hansen et al., 2004).

To the best of our knowledge, there are no studies that have introduced truthful mechanisms into the field of MARL, but it may be possible to introduce it by using agents that can learn flexibly, such as neural networks. A typical truthful mechanism is the VCG mechanism (Vickrey, 1961), which is a generalization of the pivot method used in auction theory, but whereas the subject of the report that must satisfy truthfulness must be a *valuation* (or a value function if interpreted from a RL perspective). In this study, the scope of application is different because the belief states of the environment are subject to reporting. Therefore, we introduce instead a peer prediction method (Miller et al., 2005) that guarantees truthfulness with respect to reporting beliefs about arbitrary probability distributions using *proper scoring rules* (Gneiting & Raftery, 2007).

## 3 PROBLEM DEFINITION

### 3.1 COMM-POSG

A *communicative partially-observable stochastic game* (Comm-POSG) is a class of non-cooperative Bayesian games in which every agent does not fully observe the environment but interacts each other. We define Comm-POSG as an extension of POSG (Hansen et al., 2004) with a message protocol.

**Definition 3.1** (Hansen et al., 2004) POSG $\langle n, T, \mathcal{S}, \mathcal{A}, \mathcal{X}, \mathcal{T}, \mathcal{P}, \mathcal{R} \rangle$ is a class for multi-agent decision making under uncertainty in which the state evolves over time $1 \leq t \leq T$, where

- $n$ is the number of agents,
- $T$ is a horizon $i.e.$, the episode length,
- $\mathcal{S}$ is a set of discrete/continuous state $s_t \in \mathcal{S}$ with an initial probabilistic distribution $p(s_0)$,
- $\mathcal{A}$ is a set of discrete/continuous action $a_{ti} \in \mathcal{A}$,
- $\mathcal{X}$ is a set of discrete/continuous observation $x_{ti} \in \mathcal{X}$,
- $\mathcal{T} \in \mathfrak{P}(\mathcal{S} \times \mathcal{A} \times \mathcal{S})$ is state transition probability,
- $\mathcal{P} \in \mathfrak{P}(\mathcal{S} \times \mathcal{X}^n)$ is an observation probability, and
- $\mathcal{R} : \mathcal{S} \times \mathcal{A}^n \to \mathbb{R}^n$ is a reward function that outputs an $n$-dimensional vector. □

In Comm-POSGs, every agent further follows a *message protocol* $\mathcal{Z}^{n \times n}$, where $\mathcal{Z}$ is the discrete/continuous signal space. The complete information exchanged among the agent in time is $\mathbf{Z}_t$, where $\mathbf{Z}_t := (z_{tij})_{i,j \in [\![n]\!]} \in \mathcal{Z}^{n \times n}$ is a *signal matrix* in which $(i, j)$-th entry $z_{tij}$ represents a signal from Agent $i$ to Agent $j$ at $t$. The $i$-th diagonal entry of $\mathbf{Z}_t$, $h_{ti} := z_{tii}$ represents the *pre-state*, an internal state of $i$-th agent before receiving the singals from the others. A game in Comm-POSG is denoted as $\mathcal{G} := \langle n, T, \mathcal{S}, \mathcal{A}, \mathcal{X}, \mathcal{T}, \mathcal{P}, \mathcal{R}, \mathcal{Z} \rangle$.

The objective of Comm-POSG is *social welfare* (Arrow, 1963) defined by the following,

$$J := \sum_{i=1}^{n} V^{\pi_i}; \qquad V^{\pi_i} := \mathbb{E}_{\pi_i} \left[ \sum_{t=1}^{T} \gamma^{t-1} r_{ti} \right], \tag{1}$$

where $\gamma \in [0, 1]$ is discount rate, $r_{ti}$ is reward $\pi_i$ is a stochastic policy, and $V^{\pi_i}$ is the value function.

In extensive-form games including Comm-POSG, in addition to the information in the environment, the information of other agents cannot be observed. In the optimization problem under these assumptions, a policy converges to a solution called the Bayesian Nash equilibrium (BNE) (Fudenberg, 1993). We denote the social welfare at the BNE is $J^*$, and the global maximum $\hat{J}$. In general, $J^* \neq \hat{J}$ holds, which is closely related to the information asymmetry.

### 3.2 COMMUNICATIVE RECURRENT AGENTS

In order to propose an optimization algorithm in this paper, we do not propose a concrete structure of the network, but we propose an abstract structure that can cover existing neural communication models (Sukhbaatar et al., 2016; Singh et al., 2019), namely *communicative recurrent agents* (CRAs) $\langle f_\phi, \sigma_\phi, q_\phi, \pi_\theta \rangle$, where

- $f_\phi(\hat{h}_{t-1,i}, x_{ti}) \mapsto \mathcal{Z}$ is a deep RNN for the high-dimensional input $x_{ti} \in \mathcal{X}$ and the previous post-state $\hat{h}_{t-1,i} \in \mathcal{Z}$, with a parameter $\phi$ and an initial state $\hat{h}_0 \in \mathcal{Z}$,

- $\sigma_\phi(\mathbf{z}_{ti}|h_{ti})$ is a stochastic messaging policy for a pre-state $h_{ti} := f_\phi(\hat{h}_{t-1,i}, x_{ti})$,

- $q_\phi(\hat{h}_{ti}|\hat{\mathbf{z}}_{ti})$ is a stochastic model for a *post-state* $\hat{h}_{ti} \in \mathcal{Z}$ and the received messages $\hat{\mathbf{z}}_{ti} := \mathbf{Z}_{t:i}^{\mathrm{T}} = (z_{t1i}, \ldots, z_{t,i-1,i}, h_{ti}, z_{t,i+1,i}, \ldots, z_{tni})^{\mathrm{T}}$, and

- $\pi_\theta(a_{ti}|\hat{h}_{ti})$ is the stochastic action policy with a parameter $\theta$.

These agents are trained through self-play using on-policy learning such as REINFORCE (Williams, 1992). All $n$-agents share the same weight per episode, and the weights are updated based on the cumulative reward after the episode.

In addition to the recurrent agent's output of actions with the observation series as input, the CRA has signals for communication as input and output. CRAs estimate current state of the environment and current value of the agent herself based on the post-state model with the pre-state $h_{ti}$ in the hidden layer of the RNN and the received signals $\hat{\mathbf{z}}_{ti,-i}$ from other agents. Hence, the veracity of the signals $\mathbf{z}_{ti}$ is the point of contention.

## 3.3 TRUTHFULNESS

In mechanism design, a *truthful* game (Vickrey, 1961) is a game in which all agents make an honest reporting in the Bayesian Nash equilibrium. In Comm-POSGs, the truthfulness of the game is achieved if all the sent signal equals the pre-state $z_{tij} = h_{ti}$ *i.e.,* all the agent share a complete information. In such case, every agent has the same information $\hat{\mathbf{z}}_{ti} = \mathbf{h}_{ti} := (h_{t1}, \ldots, h_{tn})^{\mathrm{T}}$ for all $i$'s and the same post-state model probability distribution, and hence the mean of the cross entropy between the distributions below will be minimized.

$$D_\phi(\mathbf{Z}_t) := \frac{1}{n}\sum_{i=1}^{n}\mathbb{H}\left[q_\phi(\hat{h}_{ti}|\hat{\mathbf{z}}_{ti})\right] + \frac{1}{n^2}\sum_{i=1}^{n}\sum_{j=1}^{n}\mathcal{D}_{\mathrm{KL}}\left(q_\phi(\hat{h}_{ti}|\hat{\mathbf{z}}_{ti}) \,||\, q_\phi(\hat{h}_{tj}|\hat{\mathbf{z}}_{tj})\right). \quad (2)$$

The first term represents the entropy of knowledge each agent has about the environment, and the second the information asymmetry between the agents. $D_\phi$ is a lower bound on the amount of true information the environment has $\mathbb{H}\left[p(s_t)\right]$. Since achieving truthfulness is essentially the same problem as minimizing $D_\phi$, it also maximizes $J^*$ simultaneously.

**Proposition 3.1.** *(global optimality) For any games $\mathcal{G}$ in Comm-POSG, if $D_\phi(\mathbf{Z}_t) = \mathbb{H}\left[p(s_t)\right]$ for $0 \leq t \leq T$ and $\mathcal{R}_i$ is symmetric for any permutation of $i \in [\![n]\!]$, $J^*(\mathcal{G}) = \hat{J}(\mathcal{G})$ holds.*

*Proof.* Let $\mathbf{w} := \langle\theta, \phi\rangle$ on a parameter space $\mathcal{W}(\mathcal{G})$, and $\mathcal{W}^*(\mathcal{G})$ the BNE. Since $J$ is obviously maximized if $\sigma_\phi$ is truthful, we prove $\sigma_\phi$ must be truthful under a given condition. To this end, we show the following Pareto-optimality.

**Lemma 3.1.** *For any $\mathcal{G}$ in Comm-POSG and given $\mathbf{w}$, if $J(\mathbf{w}) \geq J(\mathbf{w}')$ holds for any $\mathbf{w}' \in \mathcal{W}(\mathcal{G})$, then either of $U(\mathbf{w}) \geq U(\mathbf{w}')$ or $\mathcal{D}_{\mathrm{KL}}(p \,||\, q_\phi) \leq \mathcal{D}_{\mathrm{KL}}(p \,||\, q_{\phi'})$ holds, where*

$$U(\mathbf{w}) := \mathbb{E}_{q_\phi}[V^{\pi_\theta}] = \int_{s_t \in \mathcal{S}, \hat{\mathbf{z}}_t \in \mathcal{Z}^n} V^{\pi_\theta}(s_t)\,\mathrm{d}q_\phi(s_t|h_{ti}, \mathbf{z}_{t,-i})\,\mathrm{d}\sigma_\phi(\hat{\mathbf{z}}_t). \quad (3)$$

*Proof.* The first inequality $U(\mathbf{w}) \geq U(\mathbf{w}')$ indicates that $\mathbf{w}$ is on the BNE $\mathcal{W}^*(\mathcal{G})$ given $\phi$, and of the second that the belief state $q_\phi$ is as close to the true state $p$ as possible. On a fully-observable environment, from the theorem of value iteration, there exists the solution $\pi^\star(s_t)$ w.r.t. $V^{\pi^\star}(s_t)$ for any $s_t \in \mathcal{S}$. We name $\pi^\star$ and $V^\star := V^{\pi^\star}$ the *unbiased* policy and value, respectively.

Since the unbiased policy solves the objective as $J(\mathbf{w}) = n\mathbb{E}_{p(s_t)}[V^{\pi_\theta}(s_t)]$ from Eq (1), the goal intrinsicly is to find the policy $\pi^*$ as close as $\pi^\star$ for $\langle\pi^*, \phi^*\rangle \in \mathcal{W}(\mathcal{G})$, that maximizes $U(\mathbf{w})$. The $\pi^*$ further can be represented as a mixed policy made of a couple of the unbiased policy $\pi^\star$ and a *biased* policy $\pi' \neq \pi^\star$ as follows,

$$\begin{aligned}\pi^*(a_{ti}|\mathbf{x}_{ti}, \phi) &= \mathbb{E}_{q_\phi(s_{ti}|\mathbf{x}_{ti})}[\pi^\star(a_{ti}|s_{ti})] \\ &= q_\phi(s_{t0}|\mathbf{x}_{ti})\pi^\star(a_{ti}|s_{t0}) + (1 - q_\phi(s_{t0}|\mathbf{x}_{ti}))\pi'(a_{ti}|\phi, \mathbf{x}_{ti}),\end{aligned} \quad (4)$$

for the observations $\mathbf{x}_{ti} \in \mathcal{X}^n$, where $s_{t0} \in \mathcal{S}$ is the true state. Hence

$$
\begin{aligned}
V^{\pi^*}(s_{t0}|\phi) &= \mathbb{E}_{\mathcal{P}(\mathbf{x}_{ti}|s_{t0})} \left[ q_\phi(s_{t0}|\mathbf{x}_{ti}) V^\star(s_{t0}) + (1 - q_\phi(s_{t0}|\mathbf{x}_{ti})) V'(\mathbf{x}_{ti}|\phi)) \right] \\
&= q_\phi(s_{t0}) V^\star(s_{t0}) + \mathbb{E}_{\mathcal{P}(\mathbf{x}_{ti}|s_{t0})} \left[ (1 - q_\phi(s_{t0}|\mathbf{x}_{ti})) V'(\mathbf{x}_{ti}|\phi)) \right] \\
&= q_\phi(s_{t0}) V^\star(s_{t0}) + (1 - q_\phi(s_{t0})) \bar{V}'(s_{t0}|\phi),
\end{aligned}
\tag{5}
$$

where

$$
V'(\mathbf{x}_{ti}|\phi) := \int_{\mathbf{s}_t \in \mathcal{S}^{n+1}, \mathbf{a}_t \in \mathcal{A}^n} \mathcal{R}_i(s_{t0}, \mathbf{a}_t) \prod_{i=1}^n \mathrm{d}\pi'(a_{ti}|s_{ti}) q_\phi(s_{ti}|\mathbf{x}_{ti}),
\tag{6}
$$

and

$$
\bar{V}'(s_{t0}|\phi) := \mathbb{E}_{\mathcal{P}(\mathbf{x}_{ti}|s_{t0})} \left[ V'(\mathbf{x}_{ti}|\phi) \frac{1 - q_\phi(s_{t0}|\mathbf{x}_{ti})}{1 - q_\phi(s_{t0})} \right].
\tag{7}
$$

Thus, the error from the unbiased value function can be written as $V^\star(s_{t0}) - V^{\pi^*}(s_{t0}|\phi) = (1 - q_\phi(s_{t0}))(V^\star(s_{t0}) - \bar{V}'(s_{t0}|\phi))$, which is minimized if $q_\phi(s_{t0}) = 1$ as $V^\star(s_{t0}) > \bar{V}'(s_{t0}|\phi)$ by the definition. From the Jensen's equation,

$$
\begin{aligned}
\log \mathbb{E}_{p(s_{t0})} \left[ q_\phi(s_{t0}) \right] &\geq \mathbb{E}_{p(s_{t0})} \left[ \log q_\phi(s_{t0}) \right] \\
&= -\mathcal{D}_{\mathrm{KL}} \left( p \,||\, q_\phi \right) - \mathbb{H} \left[ p \right].
\end{aligned}
\tag{8}
$$

The right-hand side of the inequation corresponds to the negative cross-entropy to be maximized. Therefore, as the second term $\mathbb{H}[p]$ depends not on $\phi$, the optimization is achieved by minimizing $\mathcal{D}_{\mathrm{KL}}(p \,||\, q_\phi)$. $\square$

Suppose that $\hat{J}(\mathcal{G}) = J(\mathbf{w}) > J^*(\mathcal{G})$ for a non-truthful reporting policy s.t. $\sigma_\phi(h|h) < 1$. From lemma 3.1, $q_\phi(s_t|h_{ti})$ for an internal state $h_{ti} = f(x_{ti})$ of Agent $i$ with an encoder $f$ minimizes $\mathcal{D}_{\mathrm{KL}}(p \,||\, q_\phi(s_t|h_{ti}))$. As that $q_\phi(s_t|z_{ti}) \neq q_\phi(s_t|h_{ti})$ and $\mathcal{D}_{\mathrm{KL}}(p \,||\, q_\phi(s_t|z_{ti})) > \mathcal{D}_{\mathrm{KL}}(p \,||\, q_\phi(s_t|h_{ti}))$ contradicts the Pareto-optimality, $\sigma_\phi$ must be truthful. $\square$

## 4 PROPOSED FRAMEWORK

An obvious way to achieve truthful learning is to add $D_\phi$ as a penalty term of the objective, but there are two obstacles to this approach. One is that the new regularization term also adds a bias to the social welfare $J$, and the other is that $D_\phi$ contains the agent's internal state, post-state $\hat{h}_{ti}$, so the exact amount cannot be measured by the designer of the learner. If post-states are reported correctly, then pre-states should also be reported honestly, and thus truthfulness is achieved. Therefore, it must be assumed that the post-states cannot be observed during optimization.

Our framework, *truthful self-play* (TSP), consists of two elements: one is the introduction of *imaginary rewards*, a general framework for unbiased regularization in Comm-POSG, and the other is the introduction of peer prediction method (Miller et al., 2005), a truthful mechanism to encourage honest reporting based solely on observable variables. In the following, each of them is described separately and we clarify that the proposed framework converges to the global optimum in Comm-POSG. We show the whole procedure in Algorithm 1.

### 4.1 IMAGINARY REWARD

*Imaginary rewards* are virtual rewards passed from an agent and have a different basis $i$ than rewards passed from the environment, with the characteristic that they sum to zero. Since the most of RL environments, including Comm-POSG, are of no other entities than agent and environment, two-dimensional structure are sufficient to describe them comprehensively if we wish to distinguish the sender of the reward. To remain the social welfare of the system is real, the system must be designed so that the sum of the imaginary rewards, *i.e.,* imaginary part of the social welfare, is zero. In other words, it is not observed macroscopically and affects only the relative expected rewards of agents. The real and imaginary parts of the complex rewards are ordered by the mass parameter $\beta$ during training, which allows the weights of the network to maintain a real structure.

The whole imaginary reward is denoted as $i\mathbf{Y} = (iy_{ij})_{i,j \in [\![n]\!]}$[1] where $iy_{ij}$ is the imaginary reward passed from Agent $i$ to Agent $j$, and the complex reward for the whole game is $\mathbf{R}^+ := \mathbf{R} + i\mathbf{Y}$ where $\mathbf{R}$ is a diagonal matrix with the environmental reward $r_i$ as $(i, i)$-th entry. We write $\mathcal{G}[i\mathbf{Y}]$ as the structure in which this structure is introduced. In this case, the following proposition holds.

**Proposition 4.1.** *For any $\mathcal{G}$ in Comm-POSG, if $\mathcal{G}[i\mathbf{Y}]$ is truthful and $\mathbf{R}^+$ is an Hermitian matrix, $J^*(\mathcal{G}[i\mathbf{Y}]) = \hat{J}(\mathcal{G})$ holds.*

*Proof.* Since $\mathcal{G}[i\mathbf{Y}]$ is truthful, $J^*(\mathcal{G}[i\mathbf{Y}]) = \hat{J}(\mathcal{G}[i\mathbf{Y}])$ holds from Proposition 3.1. Further, since $\mathbf{R}^+$ is Hermitian, $iy_{ij} = -iy_{ji}$, and hence $\mathrm{Im}\,\hat{J}(\mathcal{G}[i\mathbf{Y}]) = 0$ holds; $\hat{J}(\mathcal{G}[i\mathbf{Y}]) = \hat{J}(\mathcal{G})$ holds. $\square$

This indicates that the BNE could be improved by introducing imaginary rewards: $J^*(\mathcal{G}[i\mathbf{Y}]) \geq J^*(\mathcal{G})$. Also, since $\sum_{i=1}^n \sum_{j=1}^n iy_{ij} = 0$ from the condition that $\mathbf{R}^+$ is Hermitian, the imaginary rewards affect not the social welfare of the system, which is a macroscopic objective, but only the expected rewards of each agent. The baseline in policy gradient (Williams, 1992) is an example of a function that affects not the objective when the mean gets zero. However, the baseline function is a quantity that is determined based on the value function of a single agent, whereas the imaginary reward is different in that (**1**) it affects the value function of each agent and (**2**) it is a meaningful quantity only when $n \geq 2$ and is not observed when $n = 1$.

## 4.2 PEER PREDICTION MECHANISM

The *peer prediction mechanism* (Miller et al., 2005) is derived from a mechanism design using *proper scoring rules* (Gneiting & Raftery, 2007), which aims to encourage verifiers to honestly report their beliefs by assigning a reward measure score to their responses when predicting probabilistic events. These mechanisms assume at least two agents, a reporter and a verifier.

The general scoring rule can be expressed as $\mathcal{F}(p_s \| s)$ where $p_s$ is the probability of occurrence reported by the verifier for the event $s$, and $\mathcal{F}(p_s \| s)$ is the score to be obtained if the reported event $s$ actually occurred. The scoring rule is *proper* if an honest declaration consistent with the beliefs of the verifier and the reported $p_s$ maximizes the expected value of the earned score, and it is *strictly proper* if it is the only option for maximizing the expected value. A representative example that is strictly proper is the logarithmic scoring rule $\mathcal{F}(p_s \| s) = \log p_s$, where the expected value for a 1-bit signal is the cross-entropy $p_s^* \log p_s + (1 - p_s^*) \log(1 - p_s)$ for belief $p_s^*$. One can find that $p_s = p_s^*$ is the only report that maximizes the score.

Since the proper scoring rule assumes that events $s$ are observable, it is not applicable to problems such as partial observation environments where the true value is hidden. Miller et al. (2005), who first presented a peer prediction mechanism, posed scoring to a posterior of the verifiers that are updated by the signal, rather than the event. This concept is formulated by a model that assumes that an event $s$ emits a signal $z$ stochastically and infers the type of $s$ by the signal of the reporters who receive it. The peer prediction mechanism (Miller et al., 2005) is denoted as $\mathcal{F}(p(s|z) \| s)$ under the assumption that (**1**) the type of event $s$ and the signal $z$ emitted by each type follow a given prior, (**2**) the priors are shared knowledge among verifiers, and (**3**) the posterior is updated according to the reports.

We apply the mechanism to RL, *i.e.*, the problem of predicting the agent's optimal behavior $a_{ti} \sim \pi_\theta | s_t$ for the true state $s_t \in \mathcal{S}$. In self-play, the conditions of **1** and **2** can be satisfied because the prior $\pi_\theta$ is shared among the agents, and furthermore, the post-state in Comm-POSG corresponds to **3**, so that the peer prediction mechanism can be applied to the problem of predicting agent behavior. To summarize the above discussion, we can see that we can allocate a *score matrix* $\mathcal{L}_t$ as follows,

$$\mathcal{L}_t := (\ell(a_{ti}|z_{tji}))_{i,j \in [\![n]\!]}; \qquad \ell(a_{ti}|z_{tji}) := \mathcal{F}(\pi_\theta(a_{ti}|z_{tji}) \| a_{ti}) = \log \pi_\theta(a_{ti}|z_{tji}), \qquad (9)$$

which is an $n$-order square matrix representing the score from Agent $i$ to Agent $j$.

## 4.3 THE TRUTHFUL SELF-PLAY

In TSP, a truthful system is constructed by introducing a proper scoring rule into imaginary rewards. However, since the score matrix obtained from the proper scoring rule does not satisfy Hermitianity, we perform zero-averaging by subtracting the mean of the scores from each element of the matrix,

---

[1] Note the use of $i$ for imaginary units and $i$ for indices.

---

**Algorithm 1** The truthful self-play (TSP).

---

**Require:** Comm-POSG $\mathcal{G} = \langle n, T, \mathcal{S}, \mathcal{A}^+, \mathcal{X}, \mathcal{T}, \mathcal{P}, \mathcal{R}^+ \rangle$, recurrent neural network $\langle \sigma_\phi, q_\phi, \pi_\theta \rangle$ with initial weight $\mathbf{w}_0 = \langle \theta_0, \phi_0 \rangle$ and initial state $h_0$, learning rate $\alpha > 0$, and mass parameter $\beta \geq 0$.

Initialize $\mathbf{w} \leftarrow \mathbf{w}_0$.
**for** each episode **do**
    Genesis: $s_1 \sim p(s)$,    $\hat{h}_{0i} \leftarrow h_0 \,\forall i \in [\![n]\!]$.
    **for** $t = 1$ to $T$ **do**
       1. Self-Play    Observe               $\mathbf{x}_t \sim \mathcal{P}^n(\cdot | s_t)$,

| | | |
|---|---|---|
| Update pre-state | $h_{ti} \leftarrow f_\phi(\hat{h}_{t-1,i}, x_{ti})$, | $\forall i \in [\![n]\!]$. |
| Generate message | $\mathbf{z}_{ti} \sim \sigma_\phi(\cdot | h_{ti})$, | $\forall i \in [\![n]\!]$. |
| Send message | $\mathbf{Z}_t \leftarrow (\mathbf{z}_{t1}, \ldots, \mathbf{z}_{tn})$ | |
| Receive the message | $\hat{\mathbf{z}}_{ti} \leftarrow \mathbf{Z}_{t:i}^{\mathrm{T}}$, | $\forall i \in [\![n]\!]$. |
| Update post-state | $\hat{h}_{ti} \sim q_\phi(\cdot | \hat{\mathbf{z}}_{ti})$, | $\forall i \in [\![n]\!]$. |
| Act | $a_{ti} \sim \pi_\theta(\cdot | \hat{h}_{ti})$, | $\forall i \in [\![n]\!]$. |
| Get the real reward | $\mathbf{r}_t \leftarrow \mathcal{R}(s_t, \mathbf{a}_t)$, | |

     2. Compute the score matrix with peer prediction mechanism (Miller et al., 2005),

$$\ell_{ij} \leftarrow \begin{cases} \log \pi_\theta(a_j | z_i) & (i \neq j) \\ 0 & (i = j) \end{cases} \,\forall i, j \in [\![n]\!]; \tag{12}$$

     3. Combine real and imaginary rewards and construct a complex reward.

$$\mathbf{R}_t^+ \leftarrow \mathbf{R}_t + \imath \Delta \mathcal{L}. \tag{13}$$

     4. Update the weights by policy gradient (Williams, 1992),

$$\mathbf{g}_t \leftarrow \sum_{i=1}^n r_{ti}^+ \nabla_\mathbf{w} \left[ \log \pi_\theta(a_{ti} | \hat{h}_{ti}) + \log q_\phi(\hat{h}_{ti} | \hat{\mathbf{z}}_{ti}) + \log \sigma_\phi(z_{ti} | \hat{h}_{t-1,i}, x_{ti}) \right] \tag{14}$$

$$\mathbf{w} \leftarrow \mathbf{w} + \alpha \mathrm{Re} \, \mathbf{g}_t + \alpha \beta \mathrm{Im} \, \mathbf{g}_t$$

     5. Proceed to the next state $s_{t+1} \sim \mathcal{T}(\cdot | s_t, \mathbf{a}_t)$.
    **end for**
  **end for**
  **return w**

---

thereby making the sum to zero. This can be expressed as follows using the graph Laplacian $\Delta := E - \mathbf{1}\mathbf{1}^{\mathrm{T}}/n$.

$$\mathbf{Y} = \Delta \mathcal{L}_\psi = \frac{1}{n} \begin{bmatrix} n-1 & -1 & \ldots & -1 \\ -1 & n-1 & \ldots & -1 \\ \vdots & \vdots & \ddots & \vdots \\ -1 & -1 & \ldots & n-1 \end{bmatrix} \begin{bmatrix} 0 & \ell_\psi(a_{t2}|z_{t1}) & \ldots & \ell_\psi(a_{tn}|z_{t1}) \\ \ell_\psi(a_{t1}|z_{t2}) & 0 & \ldots & \ell_\psi(a_{tn}|z_{t2}) \\ \vdots & \vdots & \ddots & \vdots \\ \ell_\psi(a_{t1}|z_{tn}) & \ell_\psi(a_{t2}|z_{tn}) & \ldots & 0 \end{bmatrix}, \tag{10}$$

to get

$$\mathbf{R}^+ = \mathbf{R} + i\Delta \mathcal{L}_\psi, \tag{11}$$

which is the formula that connects reinforcement learning and mechanism design.

We show the *truthful self-play* (TSP) in Algorithm 1. The only modification required from self-play is the imaginary reward.

**Theorem 4.1.** *(global optimality) For any $\mathcal{G}$ in Comm-POSG, TSP converges to the global optimum $\hat{J}(\mathcal{G})$ if the following convergence condition are met,*

$$\sup_\phi \left| \frac{\partial \, \mathrm{Re} \, V^{\pi_\theta}}{\partial \, \mathrm{Im} \, V^{\pi_\theta}} \right| < \beta, \tag{15}$$

Table 1: Social welfare $J$ in five Comm-POSG tasks. PP-$n$: Predator prey and TJ-$n$: Traffic junction. All the scores are negatives. The experiment was repeated thrice. The average and standard deviation are listed. **Bold** is the highest score. The models listed in the top three rows show the models optimized by SP, and the bottom row shows IC3Net optimized by TSP.

|  |  | PP-3 | PP-5 | TJ-5 | TJ-10 | TJ-20 |
|---|---|---|---|---|---|---|
| SP | LSTM | $1.92 \pm 0.35$ | $4.97 \pm 1.33$ | $22.32 \pm 1.04$ | $47.91 \pm 41.2$ | $819.97 \pm 438.7$ |
|  | CommNet | $1.54 \pm 0.33$ | $4.30 \pm 1.14$ | $6.86 \pm 6.43$ | $26.63 \pm 4.56$ | $463.91 \pm 460.8$ |
|  | IC3Net | $1.03 \pm 0.06$ | $2.44 \pm 0.18$ | $4.35 \pm 0.72$ | $17.54 \pm 6.44$ | $216.31 \pm 131.7$ |
| **TSP** | IC3Net | $\mathbf{0.69 \pm 0.14}$ | $\mathbf{2.34 \pm 0.21}$ | $\mathbf{3.93 \pm 1.46}$ | $\mathbf{12.83 \pm 2.50}$ | $\mathbf{132.60 \pm 17.91}$ |

*where $\beta < \infty$ is bounded mass parameter.*

*Proof* (in summary[2]) From Proposition A.2, $[\beta \Delta \mathcal{L}_\psi]$ is unbiased truthful. Therefore, from Proposition A.1, convergence to the global optima is achieved. □

## 5   NUMERICAL EXPERIMENT

In this section, we establish the convergence of TSP through the results of numerical experiments with deep neural nets. We consider three environments for our analysis and experiments ($\rightarrow$ Fig. 1). (**a**) a predator prey environment (PP) in which predators with limited vision look for a prey on a square grid. (**b**) a traffic junction environment (TJ) similar to Sukhbaatar et al. (2016) in which agents with limited vision learn to signal in order to avoid collisions. (**c**) StarCraft: Brood Wars (SC) explore and combat tasks which test control on multiple agents in various scenarios where agent needs to understand and decouple observations for multiple opposing units.

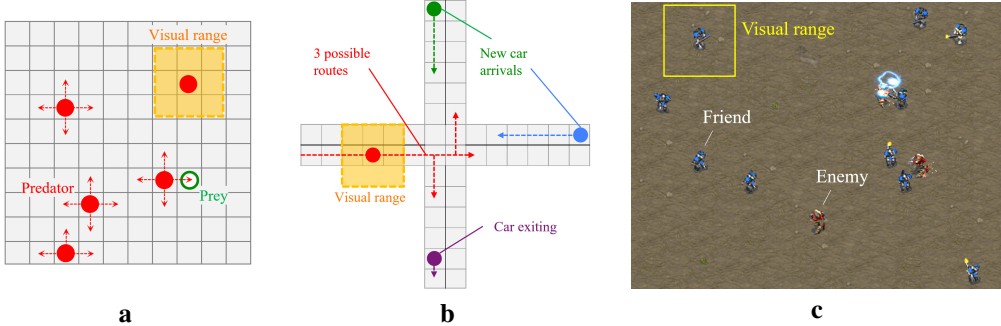

Figure 1: Experimental environments. **a**: Predator Prey (PP-$n$) (Barrett et al., 2011). Each agent receives $r_{ti} = -0.05$ at each time step. After reaching the prey, they receive $r_{ti} = 1/m$, where $m$ is the number of predators that have reached the prey. **b**: Traffic Junction (TJ-$n$) (Sukhbaatar et al., 2016; Singh et al., 2019). $n$ cars with limited sight inform the other cars of each position to avoid a collision. The cars continue to receive a reward of $r_{ti} = -0.05$ at each time as an incentive to run faster. If cars collide, each car involved in the collision will receive $r_{ti} = -1$. **c**: Combat task in StarCraft: Blood Wars (SC) (Synnaeve et al., 2016).

We compare the performances of TSP with self-play (SP) and SP with curiosity (Houthooft et al., 2016) using three tasks belonging to Comm-POSG, comprising up to 20 agents. The hyperparameters are listed in the appendix. With regard CRAs, three models namely LSTM, CommNet (Sukhbaatar et al., 2016), and IC3Net (Singh et al., 2019), were compared. The empirical mean of the social welfare function $J$ was used as a measure for the comparison. IC3Net is an improvement over CommNet, which is a continuous communication method based on LSTM. Actor-critic and value functions were added to the baselines in all the frameworks. We performed 2,000 epochs of experiment with 500 steps, each using 120 CPUs; the experiment was conducted over a period of three days.

**PP and TJ:** Table 1 lists the experimental results for each task. We can see that IC3Net with TSP outperforms the one with SP for all tasks. Fig. 2 (a) shows that TSP elicits truthful information, and

---

[2]Refer Section A for the proofs.

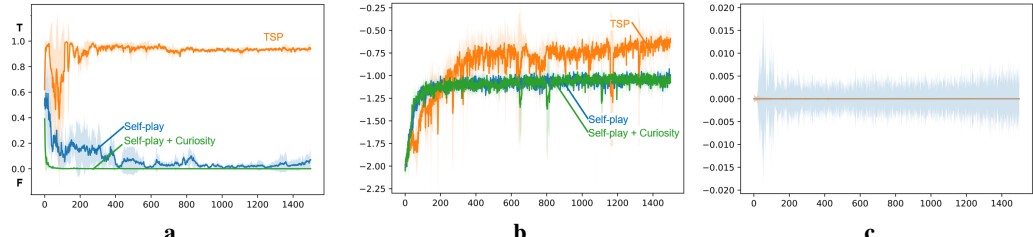

Figure 2: Learning curves in three-agent predator prey (PP-3). **a** Truthfulness, the fraction of steps that the agent shares a true pre-state ($z_{ij} = h_i$), **b** the real part of the reward, and **c** the imaginary part.

Table 2: StarCraft Results: TSP beats the baselines. The experiment was repeated thrice. All the scores are negatives. The average and standard deviation are listed. **Bold** is the highest score.

| StarCraft task | CommNet | IC3Net | IC3Net w/ TSP |
|---|---|---|---|
| Explore-10Medic $50\times50$ | $1.592 \pm 0.169$ | $2.276 \pm 1.427$ | $\mathbf{1.364 \pm 0.107}$ |
| Combat-10Mv3Ze $50\times50$ | $3.202 \pm 1.382$ | $3.815 \pm 0.392$ | $\mathbf{1.717 \pm 1.548}$ |

(b) confirms that the social welfare of TSP exceeds that of the SPs. (c) confirms that the average of the imaginary part is zero. From these experimental results, we conclude that the TSP successfully realized truthful learning and state-of-the-art in tasks comprising 3 to 20 agents.

**StarCraft:** Table 2 shows a comparison of social welfare in the exploration and combat tasks in StarCraft. (i) In the search task, 10 Medics find one enemy medic on a $50\times50$-cell grid; similar to PP, the reward is a competitive task where the reward is divided by the number of medics found. (ii) In the combat task, 10 Marines versus 3 Zealots fight on a $50\times50$ cell grid. The maximum step of the episode is set at 60. We find that IC3Net, with its information-hiding gate, performs less well than CommNet but performs better when trained in TSP due to the truthful mechanism.

## 6 CONCLUDING REMARK

Our objective was to construct a general framework for emergent unbiased state representation without any supervision. Firstly, we proposed the TSP and theoretically clarified its convergence to the global optimum in the general case. Secondly, we performed experiments involving up to 20 agents and achieved the state-of-the-art performance for all the tasks. Herein, we summarize the advantages of our framework.

1. **Strong convergence**: TSP guarantees convergence to the global optimum theoretically and experimentally; self-play cannot provide such a guarantee. Besides, the imaginary reward $i\Delta\mathcal{L}$ satisfies the baseline condition.

2. **Simple solution**: The only modification required for TSP is that $i\Delta\mathcal{L}$ should be added to the baseline in order to easily implement it for deep learning software libraries such as TensorFlow and PyTorch.

3. **Broad coverage**: TSP is a general framework, the same as self-play. Since TSP is independent of both agents and environments and supports both discrete and continuous control, it can be applied to a wide range of domains. No supervision is required.

To the best of our knowledge, introducing mechanism design to MARL is a new direction for the deep-learning community. In future work, we will consider fairness (Sen, 1984) as the social choice function. We expect that many other frameworks will be developed by using the methodology employed in this study.

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

# A  THEORY

> *"...a human brain can learn such high-level abstractions if guided by the messages produced by other humans, which act as hints or indirect supervision for these high-level abstractions; and, language and the recombination and optimization of mental concepts provide an efficient evolutionary recombination operator, and this gives rise to rapid search in the space of communicable ideas that help humans build up better high-level internal representations of their world."* (Bengio, 2012)

**Proposition A.1.** *If $[\mathcal{C}]$ is an unbiased truthful mechanism of $\mathcal{G}$, self-play with $\mathcal{G}[\mathcal{C}]$ converges to $\hat{J}(\mathcal{G})$.*

*Proof.*

Since $[\mathcal{C}]$ is unbiased, $\mathbb{E}_{\pi_\theta}[\mathcal{C}_i] = 0$ holds. Hence, for an arbitrary baseline $b$, $b + \mathcal{C}_i$ also satisfies the baseline condition. Therefore, from the policy gradient theorem (Sutton & Barto, 1998), self-play converges to $J^*(\mathcal{G}[\mathcal{C}])$. Further, since $[\mathcal{C}]$ is an unbiased truthful mechanism, $J^*(\mathcal{G}[\mathcal{C}]) = \hat{J}(\mathcal{G}[\mathcal{C}]) = \hat{J}(\mathcal{G})$ holds from Proposition 3.1. $\square$

A *general loss function* $\ell_\psi : \mathcal{A} \times \mathcal{Z} \rightarrow \mathbb{R}_\infty$ for any strictly concave nonnegative function $\psi : \mathfrak{P}(\mathcal{A}) \rightarrow \mathbb{R}_\infty$ is defined as follows:

$$\ell_\psi := \mathcal{D}_\psi \left( \pi_\theta(a_j|z_i) \| \delta(a_j|\cdot) \right), \tag{16}$$

where $\delta(a_j|\cdot)$ is a point-wise probability that satisfies $\lim_{\epsilon \to 0} \int_{B(\epsilon;\tilde{a}_j)} \mathrm{d}\delta(a_j|\tilde{a}_j) = 1$ for an open ball $B(\epsilon; \tilde{a}_j)$, and $\mathcal{D}_\psi$ is *Bregman divergence* (Bregman, 1967) defined by the following equation.

$$\mathcal{D}_\psi(p\|q) := \psi(p) - \psi(q) + \int \nabla\psi(q) \, \mathrm{d}(p - q). \tag{17}$$

Sending a truthful signal is the best response to minimize the expectation of the general loss function. For example, KL-divergence is a special case of Bregman divergence when $\psi = -\mathbb{H}[\cdot]$, and the following equation holds.

$$\mathbb{E}_{\pi_\theta}[\ell_\psi] = \int \mathcal{D}_\psi \left( \pi_\theta(a_j|z_i) \| \delta(a_j|\cdot) \right) \, \mathrm{d}\pi_\theta(a_j|h_i)$$
$$= \mathcal{D}_{\mathrm{KL}} \left( \pi_\theta(a_i|z_i) \,\|\, \pi_\theta(a_i|h_i) \right) \geq 0. \tag{18}$$

The equality holds if and only if $z_i = h_i$. Notice that $\pi_\theta(a_i|h_i) = \pi_\theta(a_j|h_i)$. Now, we generalize the zero-one mechanism to arbitrary signaling games.

**Proposition A.2.** *(Bregman mechanism) For any signaling games, if $\sup_\phi \| \mathrm{d}V^{\pi_\theta} / \mathrm{d}\beta\mathcal{I}_\psi \| < 1$, $[\mathcal{I}_\psi]$ is an unbiased truthful mechanism of $\mathcal{G} \in \mathfrak{G}$ for a general cost function:*

$$\mathcal{I}_\psi(\mathbf{a}|\mathbf{z}) := \Delta\mathcal{L}_\psi(\mathbf{a}|\mathbf{z})\mathbf{1} =$$
$$\begin{bmatrix} n-1 & -1 & \dots & -1 \\ -1 & n-1 & \dots & -1 \\ \vdots & \vdots & \ddots & \vdots \\ -1 & -1 & \dots & n-1 \end{bmatrix} \cdot \begin{bmatrix} 0 & \ell_\psi(a_2|z_1) & \dots & \ell_\psi(a_n|z_1) \\ \ell_\psi(a_1|z_2) & 0 & \dots & \ell_\psi(a_n|z_2) \\ \vdots & \vdots & \ddots & \vdots \\ \ell_\psi(a_1|z_n) & \ell_\psi(a_2|z_n) & \dots & 0 \end{bmatrix} \begin{bmatrix} 1 \\ 1 \\ \vdots \\ 1 \end{bmatrix}, \tag{19}$$

*where $\Delta := nE - \mathbf{1}\mathbf{1}^{\mathrm{T}}$ is a graph Laplacian.*

*Proof.*

The problem we dealt with is designing a scoring rule for information that satisfies two properties: 1) *regularity*, the score should be finite if the information is sufficiently correct, and 2) *properness*, the score should be maximized if and only if the information is true. The well-known example of the scoring rule is mutual information (MI), which compares a pair of probabilistic distributions $p$ and $q$ in the logarithmic domain. However, MI cannot apply to continuous actions. Instead, we introduce a more general tool, the *regular proper scoring rule* as defined below.

**Definition A.1.** (Gneiting & Raftery, 2007) *For a set $\Omega$, $\mathcal{F}(\cdot\|\cdot) : \mathfrak{P}(\Omega) \times \Omega \to \mathbb{R}_\infty$ is a* regular proper scoring rule *iff there exists a strictly concave, real-valued function $f$ on $\mathfrak{P}(\Omega)$ such that*

$$\mathcal{F}(p\|x) = f(p) - \int_\Omega f^*(p(\omega), x) \, \mathrm{d}p(\omega) + f^*(p, x) \tag{20}$$

*for $p \in \mathfrak{P}(\Omega)$ and $x \in \Omega$, where $f^*$ is a subgradient of $f$ that satisfies the following property,*

$$f(q) \geq f(p) + \int_\Omega f^*(p, \omega) \, \mathrm{d}(q - p)(\omega) \tag{21}$$

*for $q \in \mathfrak{P}(\Omega)$. We also define $\mathcal{F}$ for $q \in \mathfrak{P}(\Omega)$ as $\mathcal{F}(p\|q) := \int_\Omega \mathcal{F}(p\|x) \, \mathrm{d}q(x)$, and describe a set of regular proper scoring rules $\mathfrak{B}$.*

For $\mathcal{F}, \mathcal{F}_1, \mathcal{F}_2 \in \mathfrak{B}$, the following property holds (Gneiting & Raftery, 2007).

1. Strict concavity: if $q \neq p$, then $\mathcal{F}(q\|q) > \mathcal{F}(p\|q)$.
2. $\mathcal{F}_1(p\|q) + a\mathcal{F}_2(p\|q) + f(q) \in \mathfrak{B}$ where $a > 0$ and $f$ are not dependent on $p$.
3. $-\mathcal{D}_\psi \in \mathfrak{B}$, where $\mathcal{D}_\psi$ is the Bregman divergence.

**Lemma A.1.** *For any $\mathcal{F} \in \mathfrak{B}$ and a function $\ell_{\mathcal{F}}$ defined as shown below, if $\sup_\phi \| \mathrm{d}V_\theta^\pi / \mathrm{d}\beta\ell_\psi \| < 1$, then $[\imath\ell_{\mathcal{F}}]$ is a truthful mechanism of $\mathcal{G}$.*

$$\ell_{\mathcal{F}}(a_j|z_i) := \begin{cases} -\mathcal{F}(\pi_\theta(a_j|z_i)\|a_j) & (i \neq j) \\ 0 & (i = j) \end{cases}. \tag{22}$$

*Proof.* We prove that the surrogate objective of $\mathcal{G}[\imath\ell_{\mathcal{F}}]$, $V_\beta^{\pi_\theta} := V^{\pi_\theta} - \beta\ell_{\mathcal{F}}$ is strictly concave, and if $\nabla_\phi V_\beta^{\pi_\theta} = 0$, then $z_i = h_i$ with probability 1. We denote $\hat{\phi}$ as the *truthful parameter* where $\sigma_{\hat{\phi}}(h_i|h_i) = 1$. The policy gradient for $\phi$ is

$$\begin{aligned} \nabla_\phi V_\beta^{\pi_\theta} \, \mathrm{d}q_\phi &= \nabla_\phi V^{\pi_\theta} \, \mathrm{d}q_\phi + \beta \, \nabla_\phi \int \mathcal{F}(\pi_\theta(a_j|z_i)\|a_j) \, \mathrm{d}\pi_\theta \, \mathrm{d}q_\phi \\ &= \nabla_\phi V^{\pi_\theta} \, \mathrm{d}q_\phi + \beta \, \nabla_\phi \mathcal{F}(\pi_\theta(a_j|z_i)\|\pi_\theta(a_j|h_i)) \, \mathrm{d}q_\phi. \end{aligned} \tag{23}$$

First, we consider the local optima, i.e., $\nabla_\phi V^{\pi_\theta} \, \mathrm{d}q_\phi = \mathbf{0}$ and $\phi \neq \hat{\phi}$. For Gâteaux differential with respect to $\vec{\phi} := (\hat{\phi} - \phi)^{\mathrm{T}}/\|\hat{\phi} - \phi\|$, $\vec{\phi}\nabla V_\beta^{\pi_\theta} = \beta\vec{\phi}\nabla\ell_\psi > 0$ holds from the strict concaveity. At the global optima i.e., $\nabla_\phi V^{\pi_\theta} \, \mathrm{d}q_\phi = \mathbf{0}$ and $\phi = \hat{\phi}$, $\nabla_\phi V_\beta^{\pi_\theta} = \beta \, \nabla\ell_\psi = \mathbf{0}$ holds. Next, if $\vec{\phi}\nabla V^{\pi_\theta} < 0$, as $\sup_\phi \| \mathrm{d}V_\theta^\pi / \mathrm{d}\ell_{\mathcal{F}} \| < \beta$, the following equation holds for $\phi \neq \hat{\phi}$.

$$\begin{aligned} \vec{\phi}\nabla V_\beta^{\pi_\theta} \, \mathrm{d}q_\phi &= \vec{\phi}(\nabla V^{\pi_\theta} + \beta \, \nabla\ell_{\mathcal{F}}) \, \mathrm{d}q_\phi \\ &> \vec{\phi}\nabla V^{\pi_\theta} \, \mathrm{d}q_\phi + \sup_\phi \left\| \frac{\mathrm{d}V}{\mathrm{d}\ell_{\mathcal{F}}} \right\| \vec{\phi}\nabla\ell_{\mathcal{F}} \, \mathrm{d}q_\phi \\ &\geq \vec{\phi}\nabla V^{\pi_\theta} \, \mathrm{d}q_\phi - \inf_\phi(\vec{\phi}\nabla V^{\pi_\theta}) \, \mathrm{d}q_\phi \quad \geq 0. \end{aligned} \tag{24}$$

Hence, $\vec{\phi}\nabla V_\beta^{\pi_\theta} \geq 0$ holds, and the equality holds if and only if $\phi = \hat{\phi}$. Therefore, $V_\beta^{\pi_\theta}$ is strictly concave, and the following equation holds for $\alpha_k \in o(1/k)$.

$$\lim_{K\to\infty} \sum_{k=1}^K \frac{\nabla_\phi V_\beta^{\pi_\theta}(\phi_k)}{\| \nabla_\phi V_\beta^{\pi_\theta}(\phi_k)\|} \alpha_k = \hat{\phi}. \ (a.s.) \ \square \tag{25}$$

$\mathcal{I}_\psi$ is defined for both discrete and continuous actions. Table 3 lists examples of scoring rules $\ell_\psi$ for arbitrary actions. In particular, minimizing $\ell_\psi$ for continuous action is known as probability density estimation (Gneiting & Raftery, 2007).

$-\mathcal{I}_\psi$ is a proper scoring rule (Gneiting & Raftery, 2007) since it is a linear combination of Bregman divergence. Hence, from Lemma A.1, $[\boldsymbol{i}\mathcal{I}_\psi]$ is truthful. Besides, since $\mathbf{1}^{\mathrm{T}}\mathcal{I}_\psi = \mathbf{1}^{\mathrm{T}}\Delta\mathcal{L}_\phi\mathbf{1} = 0$, $[\boldsymbol{i}\mathcal{I}_\psi]$ is unbiased. $\square$

Table 3: Examples of the scoring rules, where $\kappa > 1, \|p\|_\kappa = (\int p(a)^\kappa \, \mathrm{d}a)^{1/\kappa}$. Readers can refer to (Gneiting & Raftery, 2007) for examples of many other functions.

| | $\mathcal{A}$ | $\mathcal{Z}$ | $\psi(p)$ | $\ell_\psi(a\mid z)$ |
|---|---|---|---|---|
| Zero-one | Discrete | Discrete | $-p$ | $-\pi_\theta(a\mid z)$ |
| Logarithmic | Discrete | Continuous | $-\mathbb{H}\,[p]$ | $-\log \pi_\theta(a\mid z)$ |
| Quadratic | Discrete | D/C | $\sum_{a \in \mathcal{A}} p(a)^2 - 1$ | $2\pi_\theta(a\mid z) - \psi(\pi_\theta(\cdot\mid z)) - 2$ |
| (Brier, 1950) | Continuous | D/C | $\|p\|_2^2$ | $2\pi_\theta(a\mid z) - \|\pi_\theta(\cdot\mid z)\|_2^2$ |
| Pseudospherical | Discrete | Continuous | $(\sum_{a \in \mathcal{A}} p(a)^\kappa)^{1/\kappa}$ | $\pi_\theta(a\mid z)^{\kappa-1}/\psi(\pi_\theta(\cdot\mid z))^{\kappa-1}$ |
| (Good, 1952) | Continuous | Continuous | $\|p\|_\kappa^{\kappa-1}$ | $\pi_\theta(a\mid z)^{\kappa-1}/\|\pi_\theta(\cdot\mid z)^\kappa\|_\kappa^{\kappa-1}$ |

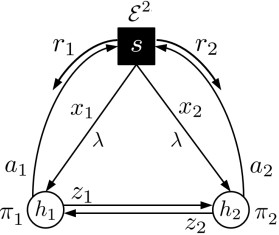 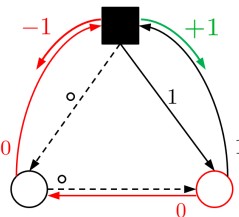

Figure 3: **Left:** One-bit two-way communication game $\mathcal{G}_{\mathrm{com}}^2$. $\mathcal{S} = \mathcal{A} = \{0,1\}, \mathcal{X} = \mathcal{S} \cup \{\circ\}$, and $p(s) = 1/2$. Agents are given correct information from the environment with a probability of $\lambda > 0$: $\mathcal{P}(s\mid s) = \lambda, \mathcal{P}(\circ\mid s) = 1 - \lambda$. $q_i(s\mid x_i, z_j)$ follows a Bernoulli distribution and estimates the state $p(s)$ of the environment from observations and signals $z_j \in \mathcal{Z} = \mathcal{X}$. Let $h_i = x_i$. **Right:** An example of the non-cooperative solutions: $s = 1, \mathbf{x} = \langle\circ, 1\rangle, \mathbf{z} = \langle\circ, 0\rangle, \mathbf{a} = \langle 0, 1\rangle, \mathbf{r} = \langle -1, 1\rangle$ and $r_1 + r_2 = 0 < 2c$.

**Theorem A.1.** *(global optimally) For any $\mathcal{G}$ in Comm-POSG, TSP converges to the global optimum $\hat{J}(\mathcal{G})$ if the following convergence condition are met,*

$$\sup_\phi \left|\frac{\partial \operatorname{Re} V^{\pi_\theta}}{\partial \operatorname{Im} V^{\pi_\theta}}\right| < \beta, \tag{26}$$

*where $\beta < \infty$ is bounded mass parameter.*

*Proof.*

From Proposition A.2, $[\iota \mathcal{I}_\psi]$ is unbiased truthful. Therefore, from Proposition A.1, convergence to the global optima is achieved. $\square$

## A.1 SELF-PLAY CONVERGES TO LOCAL OPTIMA

**Theorem A.2.** *If $\mathcal{G} \in \mathfrak{G}$ is non-truthful, self-play does not converge to the global optimum $\hat{J}(\mathcal{G})$.*

*Proof.*

**Example A.1.** *(One-bit two-way communication game) Fig. 4 shows an example of a non-cooperative partially observable environment with 1-bit state. The reward structure is presented in Table 4. The sum of rewards is maximized when both agents report the correct state to the environment,*

$$\sum_{i=1}^n \mathcal{R}_i^n(s, \mathbf{a}) = \begin{cases} 2c & (a_1 = a_2 = s) \\ 0 & (\text{otherwise}) \end{cases}.$$

*Hence, the objective varies in the range $0 \le J(\mathcal{G}_{\mathrm{com}}^2) \le 2c$.*

**Proposition A.3.** *If $c < 1, J^*(\mathcal{G}_{\mathrm{com}}^2) < \hat{J}(\mathcal{G}_{\mathrm{com}}^2)$ holds.*

*Proof.*

Table 4: $\mathcal{G}_{\text{com}}^2$'s reward function $\mathcal{R}^2(s, \mathbf{a})$. $c > 0$ is a constant that determines the nature of the game.

|       |     |           | $a_2$     |          |
| :---: | :---: | :-------: | :-------: | :------: |
|       |     |           | True      | False    |
|       |     | $(r_1, r_2)$ | $s$    | $1 - s$  |
| $a_1$ | T   | $s$       | $(c, c)$  | $(1, -1)$ |
|       | F   | $1 - s$   | $(-1, 1)$ | $(0, 0)$ |

Since $p(s) = 1/2$, we can assume $s = 1$ without loss of generality. Besides, we discuss only Agent 1 because of symmetry. From $\mathcal{Z} = \{0, 1\}$, Agent 1's messageling policy $\sigma_1$ sends the correct information $x$ or false information $1 - x$ when it knows $x$. Hence, we can represent the policy by using parameter $\phi \in [0, 1]$ as follows.

$$\sigma_\phi(z|x) = \begin{cases} \phi^z (1 - \phi)^{1-z} & (x = 1) \\ 1/2 & (x = \circ) \end{cases}, \tag{27}$$

Differentiating $\sigma_\phi$ with $\phi$ gives the following.

$$\frac{\mathrm{d}\sigma_\phi}{\mathrm{d}\phi} = \begin{cases} 2z - 1 & (x = 1) \\ 0 & (x = \circ) \end{cases} \mathcal{G}_{\text{com}}^2. \tag{28}$$

Therefore, from Eq (**??**), if $\langle \pi^*, \cdot \rangle \in \mathcal{W}^*(\mathcal{G}_{\text{com}}^2)$, then the policy gradient for $\phi$ is as follows.

$$\begin{aligned}
\frac{\mathrm{d}}{\mathrm{d}\phi} U(\pi^*, \phi) =& \frac{\mathrm{d}}{\mathrm{d}\phi} \int V_1^* \, \mathrm{d}q_1 \, \mathrm{d}\sigma_1 \, \mathrm{d}\mathcal{P} \, \mathrm{d}p = \int V_1^* \, \mathrm{d}q_1 \frac{\mathrm{d}\sigma_1}{\mathrm{d}\phi} \, \mathrm{d}\mathcal{P} \, \mathrm{d}p \\
=& \lambda \int V_1^* (2z_1 - 1) \, \mathrm{d}z_1 \Big|_{s=x_1=1} \\
=& \lambda \int (2z_1 - 1) \mathcal{R}_1 \, \mathrm{d}\pi_1^* \, \mathrm{d}\pi_2^* \, \mathrm{d}q_2 \, \mathrm{d}z_1 \, \mathrm{d}\mathcal{P} \Big|_{s=x_1=1} \\
=& \lambda(1 - \lambda) \int (2z_1 - 1) \mathcal{R}_1 \, \mathrm{d}\pi_1^* \, \mathrm{d}\pi_2^* \, \mathrm{d}q_2 \, \mathrm{d}z_1 \Big|_{s=x_1=1, x_2=\circ} \\
=& \lambda(1 - \lambda) \sum_{z_1=0}^{1} (2z_1 - 1) \mathcal{R}_1(s, \langle x_1, z_1 \rangle) \Big|_{s=x_1=1} \\
=& \lambda(1 - \lambda) \left[ \mathcal{R}_1(1, \langle 1, 1 \rangle) - \mathcal{R}_1(1, \langle 1, 0 \rangle) \right] \\
=& \lambda(1 - \lambda)(c - 1) < 0.
\end{aligned} \tag{29}$$

As the policy gradient is negative from the assumption of $c \in (0, 1)$, $\phi^* = 0$ gets the Nash equilibrium from $\phi \geq 0$, thereby resulting in always sending false information to the opposite:

$$\sigma_{\phi^*}(z|x) = \begin{cases} 1 - z & (x = 1) \\ 1/2 & (x = \circ) \end{cases}. \tag{30}$$

Let $J\langle x_1, x_2 \rangle := J|_{\mathbf{x}=\langle x_1, x_2 \rangle}$. We can get $J^*$ and $\hat{J}$ as follows.

$$\begin{aligned}
J^* =& \int J^* \langle x_1, x_2 \rangle \, \mathrm{d}\mathcal{P}^2 \, \mathrm{d}p \\
=& J^* \langle 1, 1 \rangle \lambda^2 + J^* \langle 1, \circ \rangle \cdot 2\lambda(1 - \lambda) + J^* \langle \circ, \circ \rangle (1 - \lambda)^2 \\
=& 2c\lambda^2 + 0 + 2c/2^2 \cdot (1 - \lambda)^2 \\
=& 2c \left[ \lambda^2 + \frac{1}{4}(1 - \lambda)^2 \right],
\end{aligned} \tag{31}$$

and

$$
\begin{aligned}
\hat{J} &= \int \hat{J}\langle x_1, x_2 \rangle \, \mathrm{d}\mathcal{P}^2 \, \mathrm{d}p \\
&= \hat{J}\langle 1,1 \rangle \lambda^2 + \hat{J}\langle 1, \circ \rangle \cdot 2\lambda(1-\lambda) + \hat{J}\langle \circ, \circ \rangle (1-\lambda)^2 \\
&= 2c\lambda^2 + 2c \cdot 2\lambda(1-\lambda) + 2c/2^2 \cdot (1-\lambda)^2 \\
&= 2c\left[ \lambda^2 + 2\lambda(1-\lambda) + \frac{1}{4}(1-\lambda)^2 \right],
\end{aligned}
\tag{32}
$$

respectively. Therefore, $\hat{J} - J^* = 4c\lambda(1-\lambda) > 0$, and $J^* < \hat{J}$ holds. $\square$

From Proposition A.1, $\mathcal{G} = \mathcal{G}^2_{\mathrm{com}}$ is the counterexample that global optimally does not occur. $\square$

## A.2 ZERO-ONE MECHANISM SOLVES $\mathcal{G}^2_{\mathrm{com}}$.

**Proposition A.4.** *(zero-one mechanism) Let* $\ell : \mathcal{A} \times \mathcal{Z} \to \{0,1\}$ *be a zero-one loss between an action and a message* $\ell(a_i | z_j) := a_j(1 - z_i) + (1 - a_i)z_i$, *and*

$$
\mathcal{I}(\mathbf{a}|\mathbf{z}) := \begin{bmatrix} 1 & -1 \\ -1 & 1 \end{bmatrix} \begin{bmatrix} 0 & \ell(a_2|z_1) \\ \ell(a_1|z_2) & 0 \end{bmatrix} \begin{bmatrix} 1 \\ 1 \end{bmatrix}.
\tag{33}
$$

*If* $\beta > (1-c)(1-\lambda)/\lambda$, *then* $[\imath\mathcal{I}]$ *is an unbiased truthful mechanism of* $\mathcal{G}^2_{\mathrm{com}}$, *and self-play with* $\mathcal{G}^2_{\mathrm{com}}[\imath\mathcal{I}]$ *converges to the global optima* $\hat{J}(\mathcal{G}^2_{\mathrm{com}}) = 2c[1 - 3/4 \cdot (1-\lambda)^2]$.

*Proof.*

The following equation holds.

$$
\begin{aligned}
\frac{\mathrm{d}}{\mathrm{d}\phi} V^*_{\beta,1} \, \mathrm{d}q_1 =& \frac{\mathrm{d}}{\mathrm{d}\phi} \int (V^*_1 - \beta \mathcal{I}_1 \, \mathrm{d}\pi^*_2 \, \mathrm{d}q_2) \, \mathrm{d}q_1 \\
=& - \lambda(1-\lambda)(1-c) - \beta \int \mathcal{I}_1 \, \mathrm{d}\pi^*_2 \, \mathrm{d}q_1 \, \mathrm{d}q_2 \frac{\mathrm{d}\sigma_1}{\mathrm{d}\phi} \, \mathrm{d}\sigma_2 \, \mathrm{d}\mathcal{P}^2 \, \mathrm{d}p \\
=& - \lambda(1-\lambda)(1-c) - \beta\lambda \int \mathcal{I}_1 \, \mathrm{d}\pi^*_2 \, \mathrm{d}q_2 (2z_1 - 1) \, \mathrm{d}z_1 \, \mathrm{d}\sigma_2 \, \mathrm{d}\mathcal{P} \bigg|_{s=x_1=1} \\
=& - \lambda(1-\lambda)(1-c) - \beta\lambda^2 \int (2z_1 - 1)\ell(a_2|z_1) \, \mathrm{d}\pi^*_2 \, \mathrm{d}q_2 \, \mathrm{d}z_1 \bigg|_{s=x_1=x_2=1} \\
=& - \lambda(1-\lambda)(1-c) - \beta\lambda^2 \sum_{z_1=0}^{1} (2z_1 - 1)\ell(1|z_1) \\
=& - \lambda(1-\lambda)(1-c) - \beta\lambda^2(\ell(1|1) - \ell(1|0)) \\
=& - \lambda(1-\lambda)(1-c) + \beta\lambda^2 \\
=& \lambda^2 \left[ 1 - (1-c)\frac{1-\lambda}{\lambda} \right].
\end{aligned}
\tag{34}
$$

Therefore, if $\beta > (1-c)(1-\lambda)/\lambda$, then $\phi^* = 1$ holds, and $J^* = \hat{J}$ holds. The value of $\hat{J}$ is clear from the proof of Lemma A.1. $\square$

$[\imath\mathcal{I}]$ is also known as the *peer prediction* method (Miller et al., 2005), which is inspired by peer review. This process is illustrated in Fig. 4 Left, and the state-action value functions are listed in Fig. 4 Right.

## B COMPLEXITY ANALYSIS

Although the computational complexity of $\beta\mathcal{I}_\psi$ per iteration is $O(n^3)$ as it involves the multiplication of $n$-order square matrices, we can reduce it to $O(n^2)$ to obtain $\mathcal{I}_\psi = n\mathcal{L}_\psi - \mathbf{1}^{\mathrm{T}}\mathcal{L}_\psi\mathbf{1}$. The spatial complexity is $O(n^2)$, and the sample size is $O(n)$.

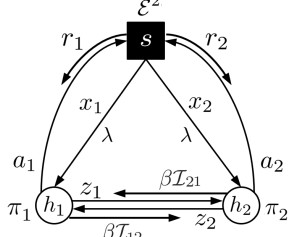

| | | $z_2$ | | |
|---|---|---|---|---|
| | | **True** | False | Unobserved |
| | T | $(c, c)$ | $(c + \beta, c - \beta)$ | $(c, c)$ |
| $z_1$ | F | $(c - \beta, c + \beta)$ | $(c, c)$ | $(1, -1)$ |
| | U | $(c, c)$ | $(-1, 1)$ | $(c/4, c/4)$ |

Figure 4: **Left:** The information released by one agent $i$ is verified by another agent, and if the information is incorrect, a penalty is given. **Right:** State-action value function of $\mathcal{G}_{\text{com}}^2[i\mathcal{I}]$: $Q_i(s_i, \langle z_i, \cdot \rangle)$.

## C    EXPERIMENTAL ENVIRONMENTS

In the experiment, we used two partial observation environments. This setting is the same as that adopted in existing studies (Sukhbaatar et al., 2016; Singh et al., 2019). Fig. 1 shows the environments.

### C.1    PREDATOR PREY (PP)

Predator-Prey (PP) is a widely used benchmark environment in MARL (Barrett et al., 2011; Sukhbaatar et al., 2016; Singh et al., 2019). Multiple predators search for prey in a randomly initialized location in the grid world with a limited field of view. The field of view of a predator is limited so that only a few blocks can be seen. Therefore, for the predator to reach the prey faster, it is necessary to inform other predators about the prey's location and the locations already searched. Thus, the prey's location is conveyed through communication among the predators, but predators can also send false messages to keep other predators away from the prey.

In this experiment, experiments are performed using PP-3 and PP-5, which have two difficulty levels. In PP-3, the range of the visual field is set to 0 in a $5 \times 5$ environment. In PP-5, the field of view is set to 1 in a $10 \times 10$ environment. The numbers represent the number of agents.

### C.2    TRAFFIC JUNCTION (TJ)

Traffic Junction (TJ) is a simplified road intersection task. An $n$ body agent with a limited field of view informs the other bodies of its location to avoid collisions. In this experiment, the difficulty level of TJ is changed to three. TJ-5 solves the task of crossing two direct one-way streets. For TJ-10, there are two lanes, and each body can not only go straight, but also turn left or right. For TJ-20, the two-lane road will comprise two parallel roads, for a total of four intersections. Each number corresponds to $n$.

In the initial state, each vehicle is given a starting point and a destination and is trained to follow a determined path as fast as possible while avoiding collisions. The agent is in each body and takes two actions, i.e., accelerator and brake, in a single time step. It is crucial to ensure that other vehicles do not approach each other to prevent collision while making good use of multiagent communication. That is similar to blinkers and brake lights.

### C.3    STARCRAFT: BLOOD WARS (SC)

**Explore:** In order to complete the exploration task, the agent must be within a specific range (field of view) of the enemy unit. Once the agent is within the enemy unit's field of view, it does not take any further action. The reward structure of the enemy unit is the same as the PP task, with the only difference being that the agent The point is that instead of being in the same place, you explore the enemy unit's range of vision and get a non-negative reward. Medic units that do not attack enemy units are used to prevent combat from interfering with the mission objective. For observation, for each agent, it is the agent's *(absolute x, absolute y)* and the enemy's *(relative x, relative y, visible)*, where *visiblea* is a visual range. If the enemy is not in exploration range, the relative x and relative y are zero. The agent has nine actions to choose from: eight basic directions and one stay action.

**Combat:** Agents make their own observations *(absolute x, absolute y, health point + shield, weapon cooldown, previous action)* and *(relative x, relative y, visible, health point + shield, weapon cooldown)*. *Relative x and y* are only observed when the enemy is visible, corresponds to a visible flag. All observations are normalized to lie between (0, 1). The agent must choose from 9+M actions, including 9 basic actions and 1 action to attack M agents. The attack action is only effective if the enemy is within the agent's view, otherwise is a no-problem. In combat, the environment doesn't compare to Starcraft's predecessors. The setup is much more difficult, restrictive, new and different, and therefore not directly comparable.

In Combat task, we give a negative reward $r_{time} = -0.01$ at each time step to avoid delaying the enemy team's detection. When an agent is not participating in a battle, at each time step, the agent is rewarded with (i) normalized health status at the current and previous time step, and (ii) normalized health status at the previous time step displays the time steps of the enemies you have attacked so far and the current time step. The final reward for each agent consists of (i) all remaining health * 3 as a negative reward and (ii) 5 * m + all remaining health * 3 as a positive reward if the agent wins. Give health*3 to all living enemies as a negative reward when you lose. In this task, a group of enemies is randomly initialized in half of the map. Thus the other half making communication-demanding tasks even more difficult.

## D  HYPERPARAMETERS

Table 5: Hyperparameters used in the experiment. $\beta$ are grid searched in space {0.1, 1, 10, 100}, and the best parameter is shown. The other parameters are not adjusted.

|  | **Notation** | **Value** |
|---|---|---|
| Agents | $n$ | $\{3, 5, 10, 20\}$ |
| Observation | $x_{ti} \in \mathcal{X}$ | $\mathbb{R}^9$ |
| Internal state | $h_{ti} \in \mathcal{H}$ | $\mathbb{R}^{64}$ |
| Message | $z_{ti} \in \mathcal{Z}$ | $\mathbb{R}^{64}$ |
| Actions | $a_{ti} \in \mathcal{A}$ | $\{\leftarrow, \uparrow, \downarrow, \rightarrow, a_{\text{stop}}\}$ |
| True state | $s_t \in \mathcal{S}$ | $\{0, 1\}^{25 \sim 400}$ |
| Episode length | $T$ | 20 |
| Learning rate | $\alpha$ | 0.001 |
| Truthful rate | $\beta$ | 10 |
| Discount rate | $\gamma$ | 1.0 |
| Metrics | $\psi$ | $-\mathbb{H}[\cdot]$ |

