# OpenReview forum: "Truthful Self-Play"
_ICLR.cc/2023/Conference — ICLR 2023 poster_

### Official Review · Reviewer_nN62 · 2022-10-24

**Confidence:** 3
**Correctness:** 4
**Technical Novelty And Significance:** 3
**Empirical Novelty And Significance:** 3
**Recommendation:** 8

**Clarity, Quality, Novelty And Reproducibility:**

Quality
- Novel idea that improves self-play through a simple modification.
- Empirical results show the effectiveness. Experimental evaluations are extensive.

Clarify
- Mostly clear. Claims are well supported. Some details missing that could be helpful for reproducibility


**Strength And Weaknesses:**

Strength
- This work is novel in the sense that it is the ﬁrst attempt to apply mechanism design to multigent evolutionary learning.
TSP’s convergence is theoretically guaranteed for arbitrary policies and environments.
- The method seems relatively easy to implement and numerical experiments show the effectiveness of TSP.

Weakness
- The authors could add a discussion on what causes self-play + curiosity’s bad performance.
- The author should at least discuss the scalability since the experiments are on simpler and smaller environments.
- The learning curves show that TSP is not as stable as baselines, how many random seeds are used in the evaluation? And what causes the instability issue?


**Summary Of The Paper:**

Paper propose a method for multi-agent reinforcement learning in non-cooperative partially observable environments with communication. The proposed method, TSP, adds imaginary rewards using the peer prediction method by evaluating the validity of information exchanged between agents. TSP has guaranteed convergence to the global optimum and has good empirical performance.


**Summary Of The Review:**

This work considers multi-agent reinforcement learning in non-cooperative POMDP. The proposed method, TSP, generalizes self-play to non-cooperative partially observable environments via mechanism design. The method has a convergence guarantee and is shown to outperform state-of-the-art performance for various multi-agent tasks.
However, there are some weaknesses that I hope the authors could address in the next version.

---

> ### Author Response · Authors · 2022-11-11
> **Response**
>
> Thank you for your sincere suggestions.
>
> > The author should at least discuss the scalability since the experiments are on simpler and smaller environments.
>
> We have given a theoretical proof and hence are confident that it will work equally well at larger scales. As for practical issues, we have already discussed the evaluation of computational complexity on Section B (p.16), but is that not enough?
>
> > The learning curves show that TSP is not as stable as baselines, how many random seeds are used in the evaluation? And what causes the instability issue?
>
> We used three random seeds per experiment.

---

### Official Review · Reviewer_N5E1 · 2022-10-24

**Confidence:** 3
**Correctness:** 3
**Technical Novelty And Significance:** 4
**Empirical Novelty And Significance:** 2
**Recommendation:** 5

**Clarity, Quality, Novelty And Reproducibility:**

Clarity:

Found the paper really unclear!
* Lots of confusing terms - is evolutionary learning any different to automatic curriculum? Just trying to understand the jargon here.
* Unclear what internal-state refers to in the introduction -> is this the internal beliefs of an agent?
* Why is this framed as faithful state representations - the question is about the truthfulness of the communication channel which is explicitly separate to the state space!

Novelty:
I can’t tell if this is Novel cause it just seems like opponent modelling and they haven’t explained why its different.


**Strength And Weaknesses:**

## Strengths
* Strong theoretical grounding
* Experiments on lots of environments
* This is useful

## Weaknesses
* The math is kinda OOT in my opinion. Section 3 could be way shorter. The use of imaginary numbers is unnecesary - simple use R^2 and then apply a normalization over a single dimension.
* (Figure a) Only showing the Real part of the reward signal?  Can we see how the second reward signal changes over time?
* Experimental results are lacking. I appreciate the variety of environments applied but simply reporting numbers isn’t convincing. I’d like to understand how messages look with some visualisation and how they change over time.
* I’d also like to understand what optimal or desired behaviour looks like in these environments - can you get to welface 0 in PP or TJ?
* Please info the reader more of your baselines (CommNET and IC3Net)
* Why didn’t you run CommNet w TSP
* Why don’t you have MAPPO as a baseline?
* The proper scoring rules seems to be no different agent modelling in other agents. Could you add some of these into the related work section just to help differentiate (or explain in rebuttal if i’m being slow)


**Summary Of The Paper:**

Truth Self Play (TSP) is a method for improving SP in partially observable environments which involve communication. The motivation is well grounded adding additional reward term to evaluate the truthfulness of communication sent. This is implemented by the additional reward  p(a_j | z_ij), which is to say, force agent j to listen to the communication sent from agent i.  In a reverse way, as the listener has to care about the message, the speaker (as this is self-play) starts providing useful information.

**Summary Of The Review:**

The paper is to a good standard and with clarity could be a very high impact paper. Theoretical justifications are great, I would like more experimental analysis and clear terminology to understand implications.

---

> ### Author Response · Authors · 2022-11-11
> **Response**
>
> Thank you for your sincere suggestions.
>
> > - The math is kinda OOT in my opinion. Section 3 could be way shorter. The use of imaginary numbers is unnecesary - simple use R^2 and then apply a normalization over a single dimension.
> > - (Figure a) Only showing the Real part of the reward signal? Can we see how the second reward signal changes over time?
>
> The reason for using imaginary numbers stems from their property of averaging to zero in a certain time step or frequency, which is a similar motivation to the use of imaginary numbers to describe alternating current relative to direct current in electromagnetism (as you know, energy is a real number, so it is not impossible to describe it in one dimension as well). We has dared to remain left in because it allows for a wider range of future considerations than simply notating them on 2 dimensions.
>
> > Please info the reader more of your baselines (CommNET and IC3Net)
>
> Self-contained is achieved by referring to previous papers. For the specific network structure, please refer to the original paper.
>
> [1] Sainbayar Sukhbaatar, Rob Fergus, et al. Learning multiagent communication with backpropagation.
> In NIPS, 2016.
>
> [2] Amanpreet Singh, Tushar Jain, and Sainbayar Sukhbaatar. Learning when to communicate at scale in
> multiagent cooperative and competitive tasks. In ICLR, 2019.
>
> > Why didn’t you run CommNet w TSP
>
> The model SOTA is IC3Net, because it is sufficient to apply it to IC3Net to show that it is better than the simple SP of the TSP with limited computational resources. The authors plan to make the source code available and these can be easily reproduced in docker. Readers wishing to do so can also follow up with CommNet.
>
> > Why don’t you have MAPPO as a baseline?
>
> Theoretically, our frameworks can apply to various optimisers including PPO. However, due to the difficulty of covering existing ones all with limited resources, we believe that an experimental understanding of the scope of application should be a future work.
>
> > Lots of confusing terms - is evolutionary learning any different to automatic curriculum? Just trying to understand the jargon here.
>
> Evolutionary learning is a method of learning due to the fact that neural networks are structures that animals have acquired evolutionarily, and tends to focus on complexity in distributed environments rather than mathematical simplicity, such as convex optimisation. Theoretical understanding of the flexibility of neural networks has been an issue at ICLR for a long time, although theoretical understanding has not caught up yet as it is something that started with the results. For example, the paper that took the best paper is one of them.
>
> [3] Zhang, Chiyuan, et al. "Understanding deep learning (still) requires rethinking generalization." Communications of the ACM 64.3 (2021): 107-115.
>
> Of course, it would be almost same as to say 'multi-agent reinforcement learning in a fully distributed environment', but evolutionary learning has nuances that are inspired by animal evolution. For example, the generative models, GANs, are also derived from evolutionary learning schemes. The automatic curriculum also implies the evolutionary approach for how knowledge is transferred between agents below. More specifically, see the following papers.
>
> [4] Yoshua Bengio. Evolving culture vs local minima. arxiv:1203.2990, 2012.
>
> > I’d also like to understand what optimal or desired behaviour looks like in these environments - can you get to welface 0 in PP or TJ?
>
> Yes.
>
> > Unclear what internal-state refers to in the introduction -> is this the internal beliefs of an agent?
>
> Yes, we mean initial beliefs.
>
> > Why is this framed as faithful state representations - the question is about the truthfulness of the communication channel which is explicitly separate to the state space!
>
> Truthfulness is the correct reporting of the knowledge an agent knows, which is assessed by the cross-entropy between the agent's internal and external knowledge, as in Eq (2). Since the basic assumption is that communication channels are used to share the state space, we assume that truthfulness is not completely separate from the state space (or state probability).

---

### Official Review · Reviewer_bEm1 · 2022-10-25

**Confidence:** 3
**Correctness:** 4
**Technical Novelty And Significance:** 3
**Empirical Novelty And Significance:** 3
**Recommendation:** 6

**Clarity, Quality, Novelty And Reproducibility:**

This paper is original and reproducible, but the quality and clarity need to be improved.

**Strength And Weaknesses:**

Strength
1. The proposed framework TSP is general and guarantees convergence to the global optimum theoretically and experimentally.

Weaknesses
1. This paper improves the self-play problem by introducing the 'truthful' reward. However, this paper lacks a full explanation of the meaning of 'truthful', in other words, we do not find relevant information about 'truthful' in the method section, and it is too farcical to use this reward as an inverse game mechanism. The contribution of this paper is too few and its innovation is general.
2. The experiment needs improvement.
   1) In Table 1, the effect of PP-5 and TJ-5 is not significantly improved; compared with IC3Net+SP, the value including standard deviation has some coincidence(need more experiments iteration). In TJ-20, there is no reasonable explanation for the reduction of effect variance.
   2) In Figure 2, the description of (a) is unclear, and the existence of (c) is unnecessary.
   3) In Table 2, whether the CommNet+TSP's performance can be further improved should be verified.
3. Spelling problems and unclear statements.
    1) Introduction para3 pertially -> partially
    2) The input of function $q_{\phi}$ is inconsistent, $\hat{h}_{t i}$ in Section3.2 and $\hat{h}_{t}$ in Section3.3.

**Summary Of The Paper:**

This paper presents a general framework named truthful self-play (TSP) which is suitable for communicative partially-observable stochastic games and analytically demonstrates convergence to the global optimum.

**Summary Of The Review:**

1. The contribution of this paper is favorable to the community .
2. The experiment needs improvement to be more convincing.
3. The language and symbolic descriptions in this paper need to be clearer to avoid misunderstandings.

---

> ### Author Response · Authors · 2022-11-11
> **Response**
>
> Thank you for your sincere suggestions.
>
> > However, this paper lacks a full explanation of the meaning of 'truthful', in other words, we do not find relevant information about 'truthful' in the method section, and it is too farcical to use this reward as an inverse game mechanism.
>
> Truthfulness is the correct reporting of the knowledge an agent knows, which is assessed by the cross-entropy between the agent's internal and external knowledge, as in Eq (2). As this is a sort of jergon in the field of game theory in microeconomics, we recommend you refer to the following key papers, which are also mentioned in this paper.
>
> [1]
> Nolan Miller, Paul Resnick, and Richard Zeckhauser. Eliciting informative feedback: The peer-prediction method. Management Science, 51(9):1359–1373, 2005.
>
> [2]
> Tilmann Gneiting and Adrian E Raftery. Strictly proper scoring rules, prediction, and estimation. Journal of the American Statistical Association, 102(477):359–378, 2007.
>
> [3]
> William Vickrey. Counterspeculation, auctions, and competitive sealed tenders. The Journal of Finance, 16(1):8–37, 1961.
>
> > In TJ-20, there is no reasonable explanation for the reduction of effect variance.
>
> In TJ-n, rewards are basically lowered by vehicles colliding with each other. Since collisions occur randomly, the more agents there are and the less information is exchanged properly, the greater the variance; the result of TJ-20 shows that many agents are able to pass smoothly and without collision, which has resulted in smaller variances.
>
> > In Figure 2, the description of (a) is unclear, and the existence of (c) is unnecessary.
>
> Figure 2 (a) reports the normalised $D_\phi$ the negative cross entropy. In terms of your opinion against (c), since there is no other part of the report that reports the imaginary part other than here, there will be one less basis for the claim that the imaginary reward does not affect the real reward should we removed the figure from. That said, further detailed suggestions to us ought to improve this.
>
> > Introduction para3 pertially -> partially
> >
> > The input of function $q_\phi$ is inconsistent, $\hat h_{ti}$ in Section 3.2 and $\hat h_{t}$ in Section3.3.
>
> We made corrections in the current revision.

---

### Public Comment · ~Michael_Noukhovitch1 · 2022-11-09
**Potentially bad-faith submission**

This submission, Truthful Self-Play, is nearly identical to previous submissions to other ML conferences dating back at least 2 years to [ICLR 2021]( https://openreview.net/forum?id=0LlujmaN0R_ ) (note this link de-anonymizes the author). There is no content difference between the submission to ICLR 2021 and this submission. It seems as though the author of this paper has submitted their work to various machine learning conferences and despite getting feedback numerous times, has not changed it at all but continued to resubmit it as-is.

This feels like bad faith on the part of the author. I understand that reviewers may vary and authors may seek to get reviewers that are more favourable to their viewpoint so I do not fault the author for resubmitting. But it does feel disrespectful to the time and effort of previous reviewers to completely ignore their reviews and change nothing between resubmissions for two whole years. For full disclosure, I was a previous reviewer of this work (but not in ICLR 2021).

I do not wish to bias the reviewers in any direction, nor is this a critique of the content of the submission itself. I would just like to draw the ACs attention to what I consider a bad faith effort in the context of our conference reviewing system.

---

> ### Author Response · Authors · 2022-11-09
> **a brief reply.**
>
> We're a bit confused. If it was really "bad-faith," we wouldn't take the time to do the experiment. The reviewer's comment in the previous conference was that they unanimously were not sure about Proposition 3.1, so we have properly added theoretical proof of this. It is very rude.
>
> It'd be violation of terms as well as being bad-faith to announce yourself as a reviewer in a peer-review system which is essentially double-blind. You should get not too emotional and talk to editors of your journal first in advance.
>
> However, we apologise in advance if, for whatever reason, we have made you feel uncomfortable. Each journal and conference has its own culture and we try to respect that to the best of our capability.

---

### Author Response · Authors · 2022-11-11
**Gratitude for Feedback**

Thank you all for your suggestions. We have given priority to answering questions and what we consider to be misunderstandings, and incorporated some of them into the paper as well. Further corrections could be made during the review period in accordance with the conference policy. If there were other critical items we overlooked, please indicate again so that we can answer them during the discussion period.

---

### Decision · Program_Chairs · 2023-01-20

**Decision:**

Accept: poster

**Justification For Why Not Higher Score:**

The generality of the work is not strong enough for a spotlight.


**Justification For Why Not Lower Score:**

It's novel and correct.

**Metareview: Summary, Strengths And Weaknesses:**

The authors present a framework for evolutionary learning to emerge an unbiased state representation without any supervision. This is relevant to multi-agent reinforcement learning in non-cooperative partially observable environmentsy. The proposed framework is a  modification of self-play that is inspired by mechanism design, so as to elicit truthful signals and make the agents cooperative. Experiments demonstrate that the proposed framework achieves state-of-the-art performance on a variety of tasks.

While there are aspects of this paper that could be improved, such as the legibility of the maths, I and the reviewers agree that it is a quality contribution. One comment mentions that the paper has been submitted and rejected previously, but this should not be held against it.

**Note From Pc:**

if the above contains the word "oral" or "spotlight" please see: "oral" presentation means -> notable-top-5% and "spotlight" means -> notable-top-25%. As stated in our emails, we are disassociating presentation type from AC recommendations